# Visual detection is locked to the internal dynamics of cortico-motor control

Alice Tomassini[1]*, Eric Maris[2], Pauline Hilt[1], Luciano Fadiga[1,3], Alessandro D'Ausilio[1,3]

1 Istituto Italiano di Tecnologia, Center for Translational Neurophysiology of Speech and Communication (CTNSC), Ferrara, Italy, 2 Radboud University, Donders Institute for Brain, Cognition and Behavior, Centre for Cognition (DCC), Nijmegen, The Netherlands, 3 Università di Ferrara, Dipartimento di Scienze Biomediche e Chirurgico Specialistiche, Ferrara, Italy

* alice.tomassini@iit.it

**Data Availability Statement:** All data files are available from the Open Science Framework Data Repository (osf.io): DOI 10.17605/OSF.IO/VSZCB

**Funding:** This work has been supported by Ministero della Salute, Ricerca Finalizzata 2016 -

## Abstract

Movements overtly sample sensory information, making sensory analysis an active-sensing process. In this study, we show that visual information sampling is not just locked to the (overt) movement dynamics but to the internal (covert) dynamics of cortico-motor control. We asked human participants to perform continuous isometric contraction while detecting unrelated and unpredictable near-threshold visual stimuli. The motor output (force) shows zero-lag coherence with brain activity (recorded via electroencephalography) in the beta-band, as previously reported. In contrast, cortical rhythms in the alpha-band systematically forerun the motor output by 200 milliseconds. Importantly, visual detection is facilitated when cortico-motor alpha (not beta) synchronization is enhanced immediately before stimulus onset, namely, at the optimal phase relationship for sensorimotor communication. These findings demonstrate an ongoing coupling between visual sampling and motor control, suggesting the operation of an internal and alpha-cycling visuomotor loop.

## Introduction

Rather than being serially ordered along distinct processing stages, action and perception are now deemed to be tightly intertwined along all of the processing stages [1]. The latter take the form of loops, whereby descending (motor) signals interact at multiple timescales with different internal (predictions) and ascending (sensory) signals that inform about the body (e.g., proprioception) and the external world (e.g., vision). Effective behavior indeed relies on a dynamic interplay between multimodal sensorimotor loops [2]. To close the sensorimotor loop, the sampling of sensory inputs must be finely synchronized to the issuing of descending commands/predictions.

Oscillatory activity is likely to coordinate the information flow across sensorimotor circuits [3–5]. Neuronal excitability is subject to ongoing oscillatory fluctuations that yield measurable consequences at both perceptual [6–9] and motor [10–12] levels. Phase synchronization of these oscillatory dynamics further provides a mechanism to appropriately time the excitability of distant groups of neurons, enabling functional coupling and selective information exchange [13–15].

Giovani Ricercatori (GR-2016-02361008) and Ministero della Salute, Ricerca Finalizzata 2018 - Giovani Ricercatori (GR-2018-12366027) to AD and by the European Union H2020 - EnTimeMent (FETPROACT-824160) to LF. The funders had no role in study design, data collection and analysis, decision to publish, or preparation of the manuscript.

**Competing interests:** The authors have declared that no competing interests exists.

**Abbreviations:** EEG, electroencephalography; EMG, electromyography; IPS, intraparietal sulcus.

Recent evidence suggests that oscillations may synchronize action onset and perceptual sensitivity. Cortical excitability [16–21] and visual performance [19, 21–26] undergo rhythmic modulations that are time-locked to movement onset (and even to the exogenous activation of the somato-motor system, [27]; see [28] for a review). Such a perceptual/neural modulation is often interpreted as reflecting a mechanism apt at optimizing active information sampling by preparing the sensory systems to the upcoming movement [29–31].

So far, perception and sensory excitability have always been probed in relation to movement onset. However, sensory information is not just "overtly" sampled at the time of movement but continuously and "covertly" integrated into the ongoing motor processing for effective planning, organization, and control of behavior.

Here we set out to investigate whether visual sensory analysis is coupled to the internal dynamics subtending continuous motor control.

To this aim, we asked human participants to perform a task requiring continuous control of the motor output without any overt effector movement, i.e., isometric force control. During this task, participants also detected unrelated visual stimuli that were shown at unpredictable times.

The ongoing relationship between the motor output (force) and brain activity recorded via electroencephalography (EEG)—classically described in the spectral domain as corticospinal coherence [32, 33]—offers a privileged window into the internal dynamics of cortico-motor control. To test our hypothesis, we thus evaluated whether visual detection performance is inherently coupled to ongoing cortico-force coherence.

## Results

We recorded EEG, electromyography (EMG), and force on 20 healthy human participants who were asked to perform 2 tasks concurrently: continuous isometric contraction and visual detection. Participants applied force with their right hand on an isometric joystick until they reached the required force level with the aid of visual feedback (see Methods for details); henceforth, they were instructed to maintain central fixation and keep tonic contraction for 5.5 seconds without feedback while waiting for the appearance of a visual dot with near-threshold contrast. The visual stimulus appeared at a random time (ranging from 1.6 to 4.6 seconds) in 85% of trials. At the end of the trial, a question mark appeared on the screen, which signaled participants to release the force and report verbally whether they had seen or not seen the stimulus (Fig 1A and 1B).

### Behavioral performance

By design, performance in the visual task is at threshold (hits: "yes" responses for stimulus-present trials, 47.43 ± 5%; misses: "no" responses for stimulus-present trials, 52.57 ± 5%; mean ± SD) and shows a low rate of false alarms ("yes" responses for stimulus-absent trials: 1.3 ± 1.7%).

To investigate whether performance in the perceptual and motor task covaried, we analyzed the force time course separately for hits and misses. Fig 1C shows the time course of the force in the pre- (left) and poststimulus (right) epochs. Overall, prestimulus force is approximately 8% lower than the instructed force level ($t_{19}$ = −3.3565, $p$ = 0.0033) and shows a slowly declining temporal trend (mean slope: −2.27% per second; $t_{19}$ = −3.1017, $p$ = 0.0059). However, produced force is not at any time point different between hits and misses (Fig 1C, left). Many metrics describing force level and variability [including absolute (error; $p$ = 0.5061) and relative (deviation; $p$ = 0.1345) difference from target force, slope ($p$ = 0.0547) and its variability ($p$ = 0.1432), as well as intertrial ($p$ = 0.8271) and within-trial force variability ($p$ = 0.1570)] are

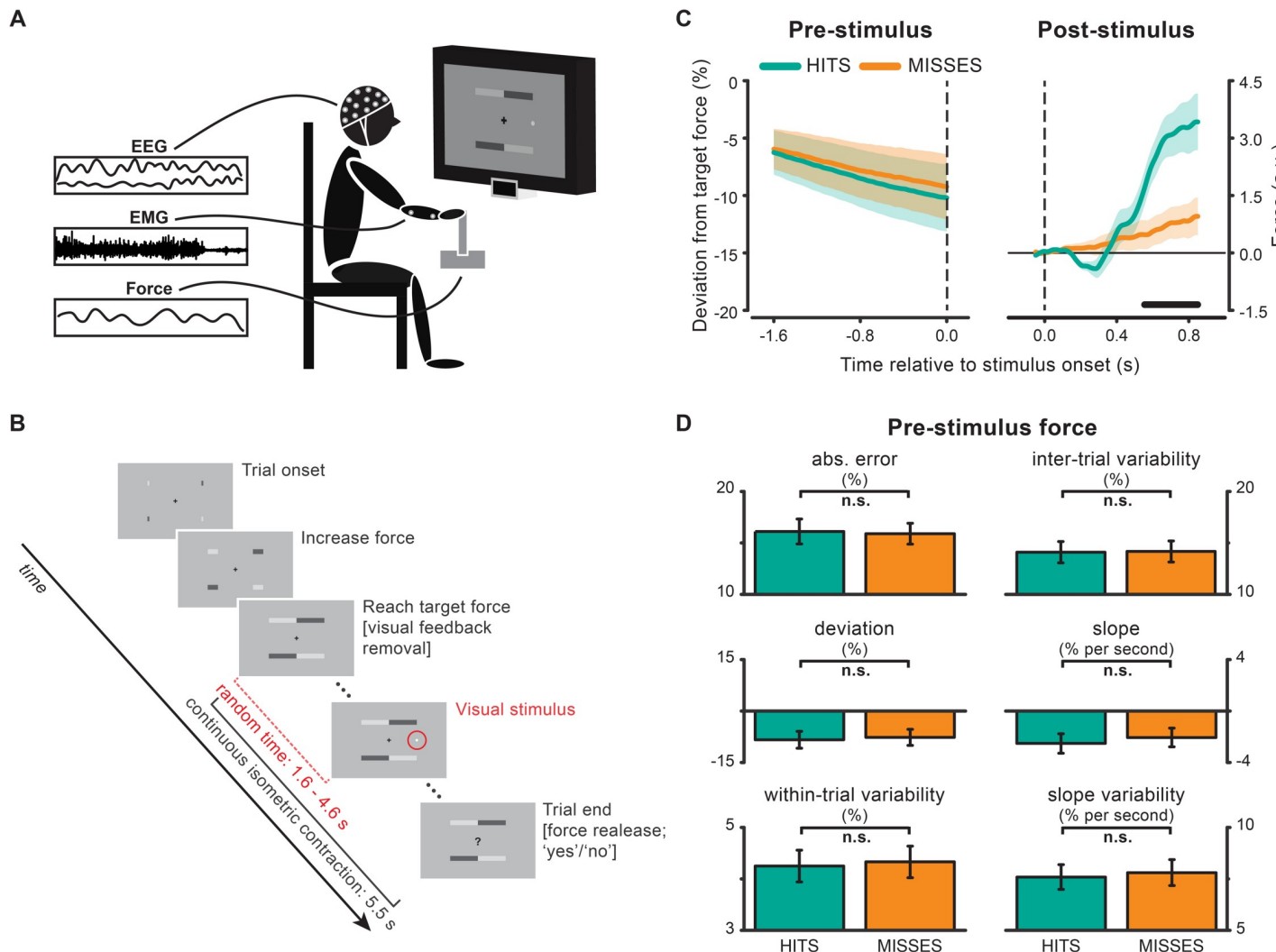

**Fig 1. Experimental setup, procedure, and behavioral results.** (A) EEG, EMG, and force were recorded while participants performed 2 tasks concurrently: visual detection and right wrist abduction to push an isometric joystick's handle towards one's own body. (B) Visual feedback of the force (4 horizontal bars elongating towards the center of the screen) was provided until participants reached the target force level (see Methods). Afterwards, participants were required to fixate and maintain stable contraction for 5.5 seconds without visual feedback. During continuous contraction, a near-threshold visual dot could appear 7.5° to the right of fixation and at a random time between 1.6 and 4.6 seconds (no stimulus was presented in 15% of trials). Trial end was signaled by a question mark prompting participants to release the contraction and report verbally whether they had seen or not seen the visual stimulus. (C) Force time courses in the pre- (left) and post- (right) stimulus period for hits and misses trials. Shaded areas represent ± 1 SEM. The black horizontal line indicates the time interval (0.55–0.85 seconds) belonging to the cluster that survived cluster-based permutation statistics for the hits–misses contrast. (D) Motor performance in the −1.6- to 0-second window before stimulus presentation quantified as absolute (error) and relative (deviation) percentage difference from target force, intertrial and within-trial force variability, slope, and slope variability. Error bars represent ± 1 SEM. For the underlying data, see https://osf.io/VSZCB/. a.u., arbitrary units; EEG, electroencephalography; EMG, electromyography; n.s., non-significant.

comparable for hits- and misses-trials (Fig 1D). Therefore, un-/successful visual performance (misses/hits) is not systematically associated with changes in force output and/or motor performance accuracy (i.e., deviation from target force).

Unlike the prestimulus epoch, force in the poststimulus epoch is modulated in a detection-dependent fashion (Fig 1C, right). Specifically, the visual stimulus evokes a biphasic response with an initial negative deflection (from approximately 0.2 to 0.4 seconds) which is visible only in hits trials, followed by a later increase in force (from approximately 0.4 seconds onwards), which is significantly larger for hits compared to misses ($p = 0.0174$; permutation test

corrected for multiple comparisons across time; cluster time interval: 0.55–0.85 seconds). These responses resemble the event-related modulations of force shown previously by Novembre and colleagues [34] for suprathreshold somatosensory and auditory stimuli and will not be investigated further in the current study.

The main analyses reported next were performed on the epoch preceding stimulus presentation time (see Fig 2 for a schematic).

## Prestimulus cortico-force coherence

Our main aim was to investigate whether visual processing is coupled to the internal motor dynamics. To provide a window into the internal motor control processes, we investigated the relationship between the produced force (the outcome of the motor control processes) and the neural activity as measured using the EEG.

The force produced during continuous isometric contraction shows a distinctive rhythmic signature centered at approximately 11 Hz (Fig 3A left), which is commonly designated as physiological tremor [35].

First, we examined whether this peripheral rhythm in the force is related to a central rhythm by computing phase coherence over the entire prestimulus window (see Methods and Fig 2A). Fig 3A (middle) shows that the cortico-force coherence spectrum has a peak around 10 Hz. Coherence in the alpha-band is spatially confined to the contralateral side and reaches maximal values at centroparietal electrodes (CP5, C5; Fig 3A, topography on the top; coherence spatially z-scored and averaged over subjects and frequencies between 9 and 11 Hz).

Coherence between peripheral (especially muscular) and cortical activity has been most commonly reported in the beta-band (see [32, 36, 37] for reviews) and much less frequently in the alpha-band [38–39]. Although the force signal does not display a distinct beta-band spectral peak (Fig 3A, left), we do observe a consistent increase in cortico-force coherence also in this frequency range ([40]; Fig 3A, middle). Though partly overlapping, the topography for beta coherence is distinct from that for alpha coherence, as it peaks over more medial and anterior electrodes (C1, C3; Fig 3A, topography on the bottom; 20–30 Hz), in full agreement with what is typically observed for cortico-muscular coherence [14, 41]. This suggests that at least partially segregated neuronal populations are involved in alpha and beta coherence.

We next assessed whether hits and misses could be differentiated based on the prestimulus cortico-motor state as evaluated through cortico-force coherence. Alpha-band fluctuations in the force are more consistently synchronized with central rhythmic activity before hits compared with misses. Specifically, hits are preceded by stronger cortico-force coherence over a large array of central and posterior electrodes (cluster $p = 0.005$; cluster frequency interval: 8.5–11.5 Hz; corrected for frequencies between 5 and 35 Hz; Fig 3B). This modulation is exclusive to the alpha range as no difference is observed within the beta range or at any other tested frequency. Moreover, it cannot be explained by differences in alpha tremor amplitude, which is comparable irrespective of the perceptual outcome (Fig 3B, left; $p > 0.05$). The effectiveness of alpha phase synchronization within the motor system is thus relevant for visual perception (in the absence of changes in the force output; see Fig 1C and 1D).

## Lagged cortico-force coherence

From a mechanistic point of view, one important aspect of the cortico-muscular/force coherence is the directionality of the interaction between cortical activity and motor output. A possible way to investigate this directionality is to look at the dependence of coherence on the relative timing (lag) between signals. We thus investigated cortico-force coherence by systematically varying the lag between the force and the EEG signal. We computed this lagged

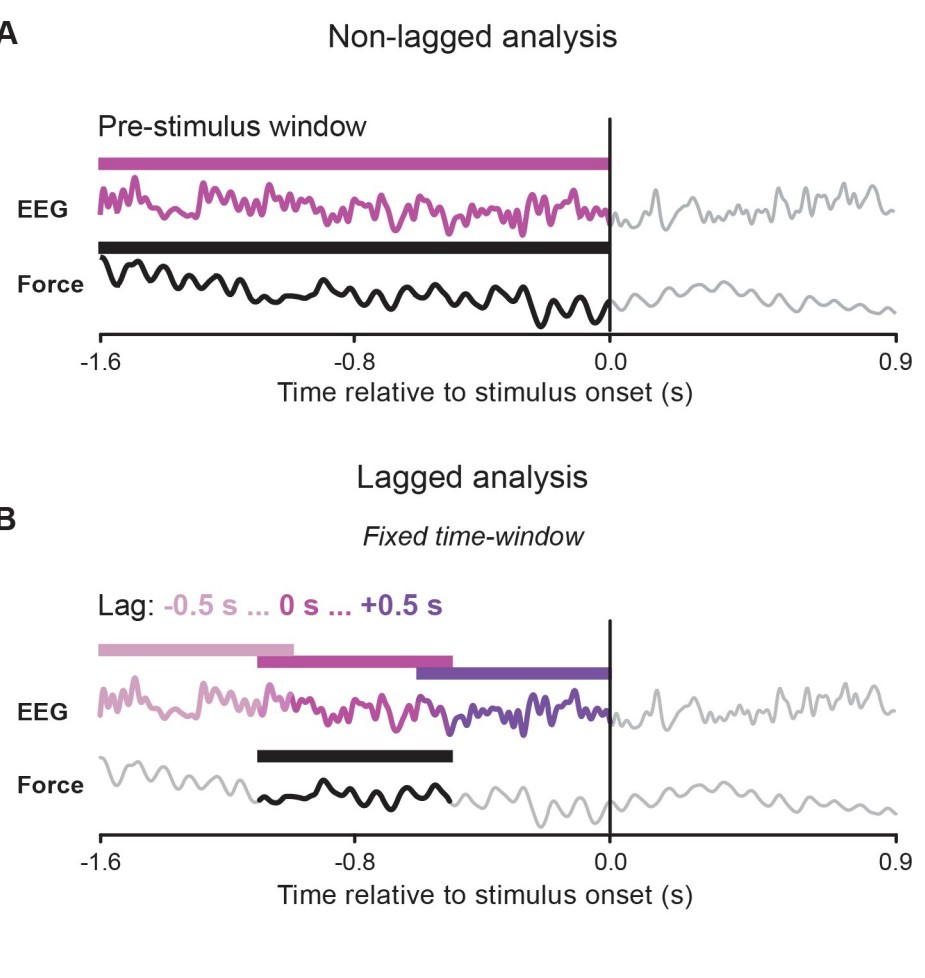

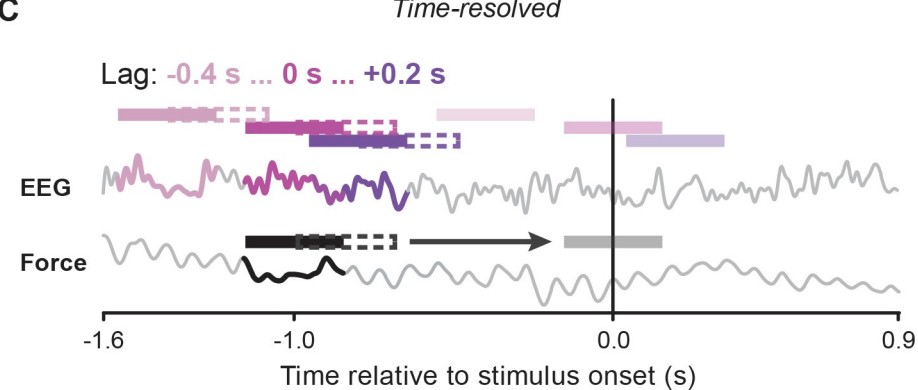

**Fig 2. Schematic illustration of the main analyses.** All panels show example force and EEG signals over the entire available epoch [time-locked to stimulus onset: from −1.6 to 0.9 seconds]. Different panels illustrate how the windowing and time-shifting (if applicable) of the signals has been performed for the main analyses. (A) Nonlagged analysis: coherence and Granger causality are computed between force (black) and EEG (violet) data windows encompassing the entire prestimulus epoch (i.e., from −1.6 to 0 seconds). (B) Lagged analysis–fixed time window: coherence is computed between a fixed 0.6-second force window (black) centered at −0.8 seconds [extending from −1.1 to −0.5 seconds] and 0.6-second EEG data windows that are either time-aligned with the force (violet; lag: 0 seconds) or shifted in time (in 10-millisecond steps) by up to 0.5 seconds in the backward (pink; lag: −0.5 seconds) and forward (dark violet; lag: +0.5 seconds) directions. (C) Lagged analysis–time-resolved: coherence is computed between 0.3-second force windows that are advanced over time (in 10-millisecond steps) from −1 second (black) up to a variable time point depending on the analysis (gray; example time is 0 seconds for illustrative purposes; see Methods for more details) and corresponding 0.3-second EEG data windows that are either time-aligned with the force (violet;

lag: 0 seconds) or shifted in time (in 10-millisecond steps) by up to 0.4 seconds in the backward (pink; lag: −0.4 seconds) and 0.2 seconds in the forward (dark violet; lag: +0.2 seconds) direction. EEG, electroencephalography.

coherence between a fixed 0.6-second force segment (from −1.1 to −0.5 seconds relative to stimulus onset) and 0.6-second EEG segments that were time-shifted (relative to the force signal) by lags ranging from −0.5 seconds (EEG precedes force) to +0.5 seconds (EEG follows force) in steps of 10 milliseconds (see Fig 2B). Fig 4A shows the lag-frequency representation

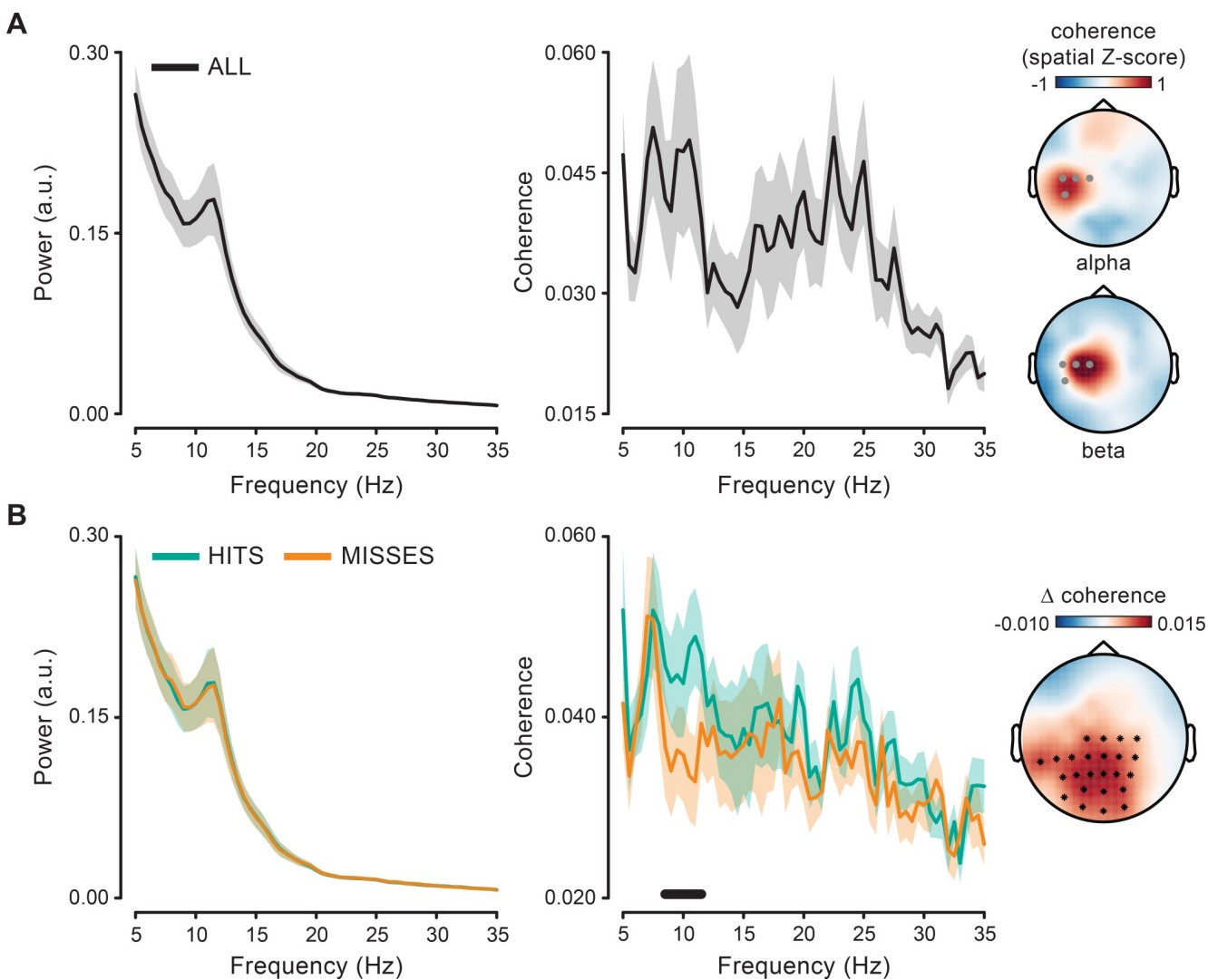

**Fig 3. Spectral content of prestimulus force and coherence with cortical activity.** (A) Force power (left) and coherence (middle) spectrum with contralateral centroparietal EEG electrodes (C1, C3, C5, CP5; marked in gray in the topographic maps) computed on the prestimulus window (−1.6 to 0 seconds). Topographies show coherence in the alpha (9–11 Hz; top) and beta (20–30 Hz; bottom) range. Coherence has been spatially z-scored before averaging across subjects and frequencies by subtracting the individual mean coherence over the electrodes and dividing the result by the standard deviation across the electrodes. (B) Same as in (A) but computed separately for hits- and misses-trials (left, middle). The black horizontal line indicates the frequency interval (8.5–11.5 Hz) belonging to the cluster that survived cluster-based permutation statistics for the hits–misses contrast. Coherence spectra are averaged over the EEG electrodes belonging to the same cluster (evaluated at 10.5 Hz; see black asterisks in the topographic map). Topography shows the hits–misses difference in coherence averaged over the cluster frequency interval (8.5–11.5 Hz). For the underlying data, see https://osf.io/VSZCB/. a.u., arbitrary units; EEG, electroencephalography.

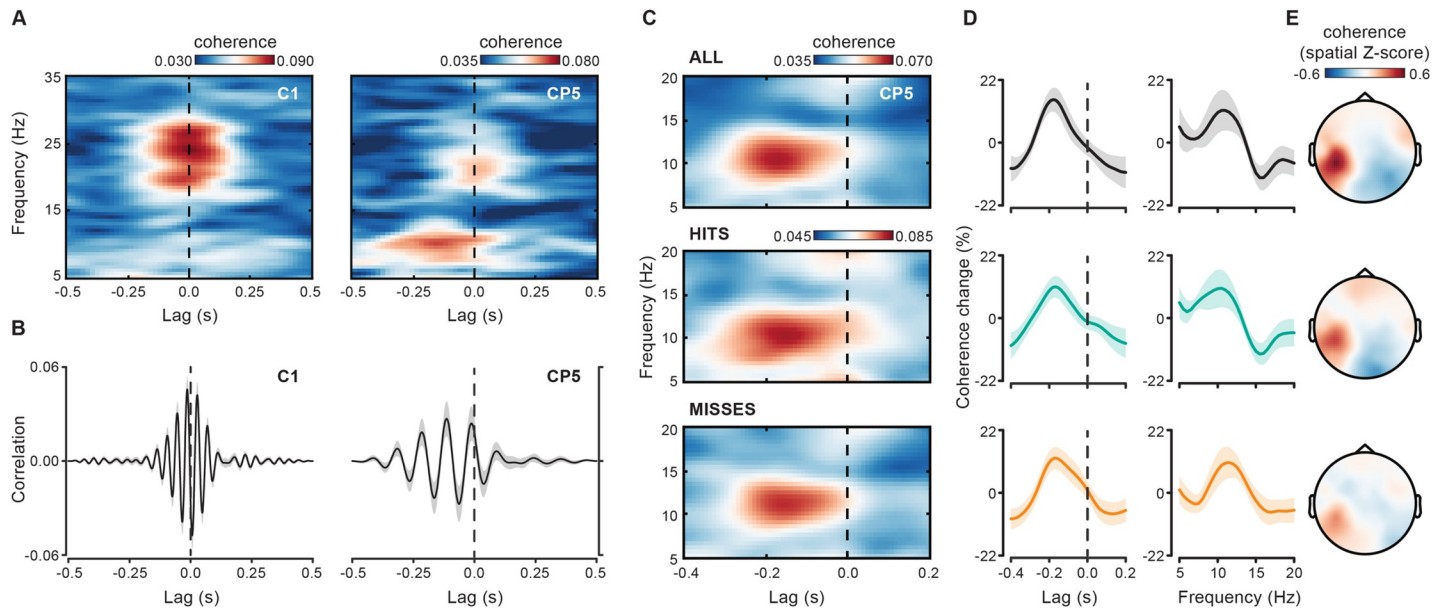

**Fig 4. Lag-dependency of cortico-force coherence.** (A) Lag- and frequency-resolved cortico-force coherence is shown for 2 EEG electrodes, C1 (left) and CP5 (right), in which beta- and alpha-band coherence is maximal, respectively. Coherence has been calculated on 0.6-second data windows (from −1.1 to −0.5 seconds) by shifting the EEG signal (relative to the force signal) by a variable amount of time (negative lags: EEG precedes force; positive lags: EEG follows force). (B) Cross-correlation between force and EEG signals [over the same electrodes as shown in (A)] that were both previously band-pass filtered (zero-phase filtering by 2-pass Butterworth, second order) in the beta (20–30 Hz; left) and alpha (8–12 Hz; right) range. Cross-correlations are normalized so that the autocorrelations at zero lag are identically 1. (C) Lag-frequency coherence representation as in (A) but computed on shorter (0.3-second) sliding data windows and then averaged over the prestimulus period for all trials as well as separately for hits- and misses-trials. (D) Lag (left) and spectral (right) tuning of cortico-force coherence expressed as the relative percentage change in coherence averaged over frequencies between 8 and 12 Hz and lags between −0.36 and 0 seconds (i.e., lag of max. alpha coherence on all trials [−0.18 seconds] ±1 SD across subjects), respectively. (E) Topographies show coherence at frequency 10.5 Hz and lag −0.2 seconds for all trials (top), hits (middle) and misses (bottom). For the underlying data, see https://osf.io/VSZCB/. EEG, electroencephalography.

of cortico-force coherence for 2 selected electrodes that better capture coherence either in the beta- (C1; left) or in the alpha- (CP5; right) band.

Whereas coherence in the beta-band is concentrated around lag zero, coherence in the alpha-band is clearly biased towards negative lags, reaching maximal values in the lag interval between −0.2 and −0.15 seconds (see S1 Fig for similar results on cortico-EMG coherence).

This peculiar lag-dependency profile is also visible in the cross-correlation functions between 1-second force and EEG segments (from −1 to 0 seconds) that were previously band-pass filtered in the relevant frequency ranges (beta-band: 20–30 Hz; alpha-band: 8–12 Hz; 2-pass Butterworth filter, second order for each single pass). Fig 4B shows that correlation between the beta-band filtered signals is nearly symmetrical around zero lag (with a very small leftward shift of few milliseconds, left). Conversely, the correlation between the alpha-band filtered signals is clearly asymmetrical and leftward-shifted with respect to lag zero (right). These results indicate that the EEG alpha rhythm foreruns a corresponding peripheral rhythm in the force by about 0.2 seconds.

To increase the temporal resolution of our analyses, we computed again lagged coherence but now using a shorter 0.3-second sliding time window that was advanced over the prestimulus epoch (from −1 to −0.35 seconds) in 10-millisecond steps (see Fig 2C). Moreover, to evaluate the consistency in the properties of alpha-band coherence across different trial categories, we performed the same analysis for all trials as well as separately for hits and misses.

Fig 4C shows the results of this analysis collapsed over the time dimension (i.e., averaged over all the prestimulus 0.3-second windows), restricted to the frequency range between 5 and

20 Hz and to the lag range between −0.4 and +0.2 seconds (see S2 Fig for comparable analyses on beta coherence). Coherence (evaluated at electrode CP5) shows selective spectro-temporal properties, being higher in the alpha-band than in the other frequencies and higher at negative than at positive lags. In particular, the lag-dependency profile peaks at −0.18 seconds for all trials and at a very similar lag for hits and misses (−0.16 seconds) (Fig 4D, left; relative % change in coherence averaged for frequencies between 8 and 12 Hz). The spectral profile is also comparable across trials subsets with maximal coherence values observed at 10.5 Hz for all trials, 10.5 Hz for hits and 11.5 Hz for misses (Fig 4D, right; relative % change in coherence averaged in the lag interval between −0.36 and 0 seconds, corresponding to the lag of maximal coherence on all trials ±1 SD across subjects). Finally, the scalp topography of alpha-band coherence is very similar across hits and misses and closely resembles the topography already obtained by computing (nonlagged) coherence over a much longer time window (1.6 seconds; see Fig 3A).

The observed increase in alpha coherence at negative lags (i.e., for EEG preceding force) is highly suggestive of a cortical alpha rhythm that drives corresponding oscillations in the motor output. In support of this, we also computed Granger causality—an established metric to estimate directed connectivity [42]—on the entire prestimulus epoch (1.6 seconds; see Fig 2A). As shown in Fig 5A, Granger connectivity from the EEG to the force, but not in the opposite direction (from the force to the EEG), shows a clear peak in the alpha range. Moreover, the topography for the EEG-to-force connectivity is very similar to that for the coherence coefficient (Fig 3A, right).

Remarkably, Granger alpha connectivity is significantly stronger for hits than for misses only in the EEG-to-force direction (cluster $p$ = 0.0065; cluster frequency interval = 9–12 Hz; corrected for frequencies between 8 and 12 Hz; Fig 5B). These results strongly confirm that alpha cortical activity predicts ("Granger causes") alpha fluctuations in the force output and this connectivity is enhanced before successful visual detection.

## Time-dependent changes in the perceptual relevance of alpha coherence

We next applied a time-resolved approach (see Fig 2C) to investigate whether the contribution of alpha-band cortico-motor connectivity to perceptual performance varies over the prestimulus window. Indeed, if the ongoing state of cortico-force coupling is relevant for perceptual processes, we could expect its impact to be maximal as we get closest to the stimulus presentation time.

Fig 6 shows the time- and lag-resolved coherence calculated for both hits and misses at 10.5 Hz (i.e., the frequency of maximal coherence for all trials combined; see Results).

Cortico-force coupling is strongly enhanced just before the onset time of seen stimuli compared with unseen stimuli. This enhancement is selectively observed at negative lags (approximately −0.2 seconds) that match well the lag tuning profile of alpha coherence (see Fig 4C and 4D). Cluster-based permutation statistics on prestimulus cortico-force coherence confirms a significant difference between hits and misses, which is concentrated over left centroparietal electrodes, at times immediately preceding stimulus onset and for EEG-leading lags ($p$ = 0.0192; corrected for multiple comparisons across time [−1 to 0 seconds] and lags [−0.4 to 0 seconds]; cluster time interval: −0.2 to 0 seconds; lag interval: −0.37 to −0.06 seconds; Fig 6).

Because of the time-resolved approach, the peri-stimulus coherence estimates are based on EEG/force data windows, which variably (depending on time and lag) embed poststimulus signals (see Fig 2C). This raises the possibility that the difference in coherence observed close to stimulus onset might reflect detection-related modulations of the responses evoked by seen (hits) and unseen (misses) stimuli (see Fig 1C for the stimulus-evoked responses in the force). To corroborate the ongoing, rather than evoked, nature of the present phenomenon, we

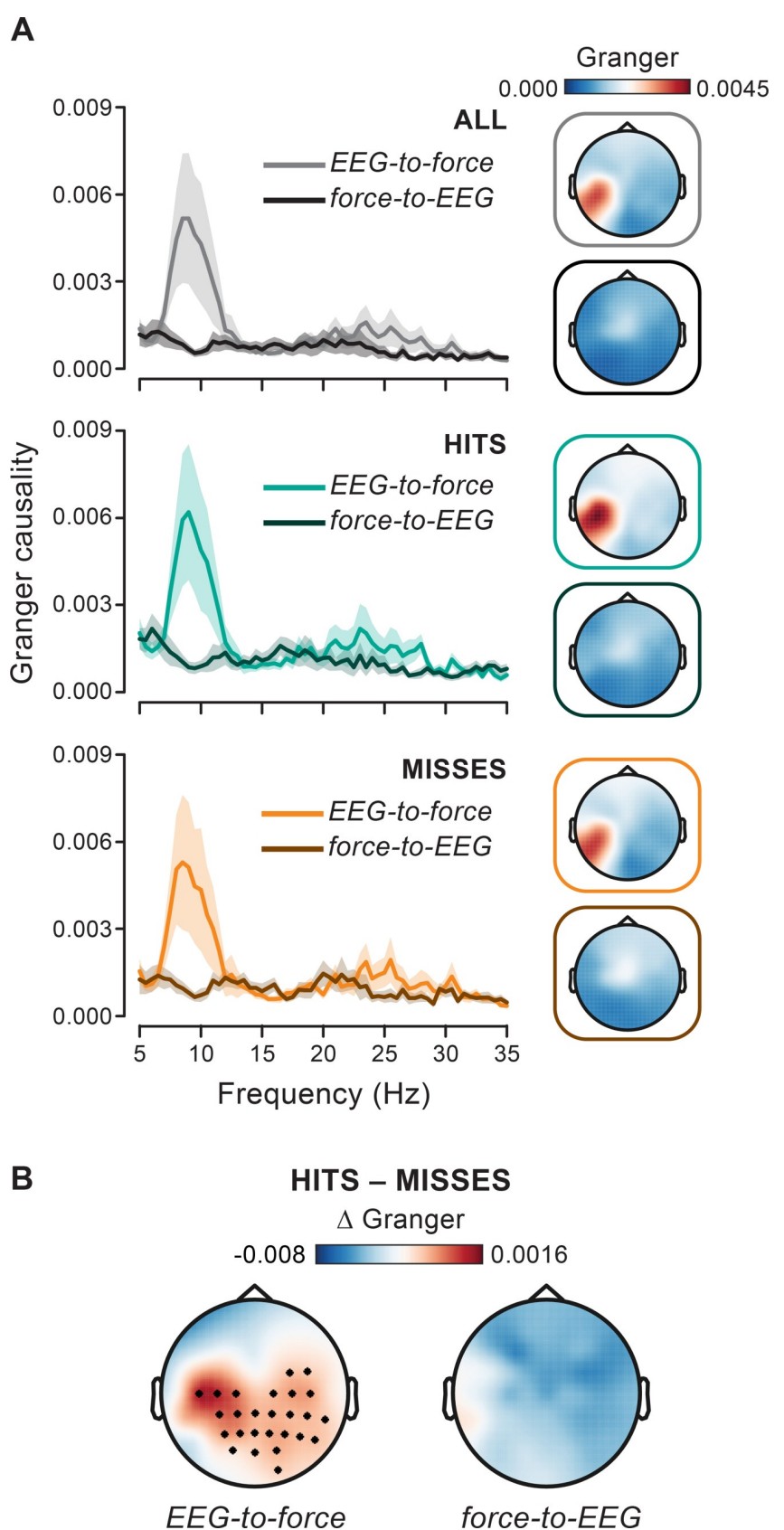

**Fig 5. Cortical alpha drives alpha fluctuations in the force: Granger causality.** (A) Granger causality in the EEG-to-force and force-to-EEG directions (evaluated at electrode CP5) computed on the entire prestimulus interval (−1.6 to 0 seconds) for all trials (top), hits (middle), and misses (bottom). Topographies show Granger causality in both directions (top: EEG-to-force; bottom: force-to-EEG). (B) Topographies show the hits-misses difference in Granger causality (left: EEG-to-force; right: force-to-EEG) evaluated at frequency 10.5 Hz (black asterisks mark electrodes belonging to the cluster that survived cluster-based permutation statistics for the hits–misses contrast). For the underlying data, see https://osf.io/VSZCB/. EEG, electroencephalography.

therefore restricted the statistical evaluation of coherence to the latest (closest to stimulus onset) time point, which encompasses uniquely prestimulus data. As shown in the bar plot of Fig 6, coherence computed between a 3-cycle (10.5-Hz) force window centered at time point −0.16 seconds (i.e., extending from approximately −0.31 to −0.01 seconds) and a corresponding EEG window (for electrode CP5) shifted in time by −0.2-second lag (i.e., extending from approximately −0.51 to −0.21 seconds) is significantly stronger for hits compared with misses (cluster $p = 0.0078$). This excludes any confound due to poststimulus data contamination.

The prestimulus nature of the observed modulation and its temporal evolution are further corroborated by Granger connectivity analyses over (nonoverlapping) 0.5-second prestimulus windows (Fig 7). A significant difference in the EEG-to-force (but, again, not in the force-to-EEG) connectivity between hits and misses is only observed in the epoch immediately preceding stimulus onset and not farther away from the stimulus.

Importantly, we checked that detection-related changes in prestimulus oscillatory power could not account for the observed changes in coherence. There are 2 main ways through which power differences between conditions could lead to spurious differences in coherence: (1) Higher power of the relevant cortical alpha activity coupled to the force can increase the signal-to-noise ratio and thus yield higher coherence estimates; (2) higher power of irrelevant cortical alpha activity that is not coupled to the force can obscure the relevant activity, decreasing the signal-to-noise ratio and thus yielding lower coherence estimates. In line with previous

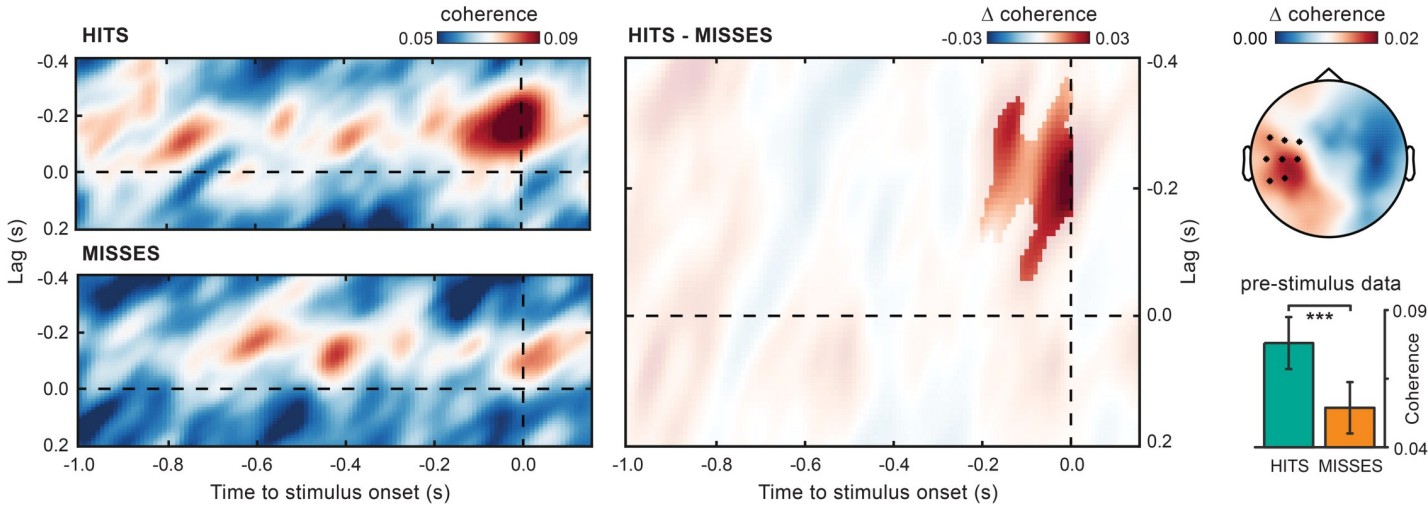

**Fig 6. Cortico-force alpha coherence just before stimulus onset predicts perception.** Lag- and time-resolved cortico-force alpha (10.5 Hz) coherence over the prestimulus period for hits, misses, and their difference (hits–misses). The highlighted area indicates the time and lag intervals belonging to the cluster that survived cluster-based permutation statistics for the hits–misses contrast. Alpha coherence is averaged over the EEG electrodes belonging to the same cluster (evaluated at time 0 seconds and lag −0.2 seconds; see black asterisks in the topographic map). The topography shows the hits–misses difference in alpha coherence averaged over the time and lag intervals belonging to the same cluster. The bar plot shows alpha coherence for the electrode CP5, calculated at lag −0.2 seconds and time −0.16 seconds, i.e., the time point closest to stimulus onset in which the analyzed data windows do not include any poststimulus data point. Error bars indicate ± 1 SEM. ***$p < 0.001$. For the underlying data, see https://osf.io/VSZCB/. EEG, electroencephalography.

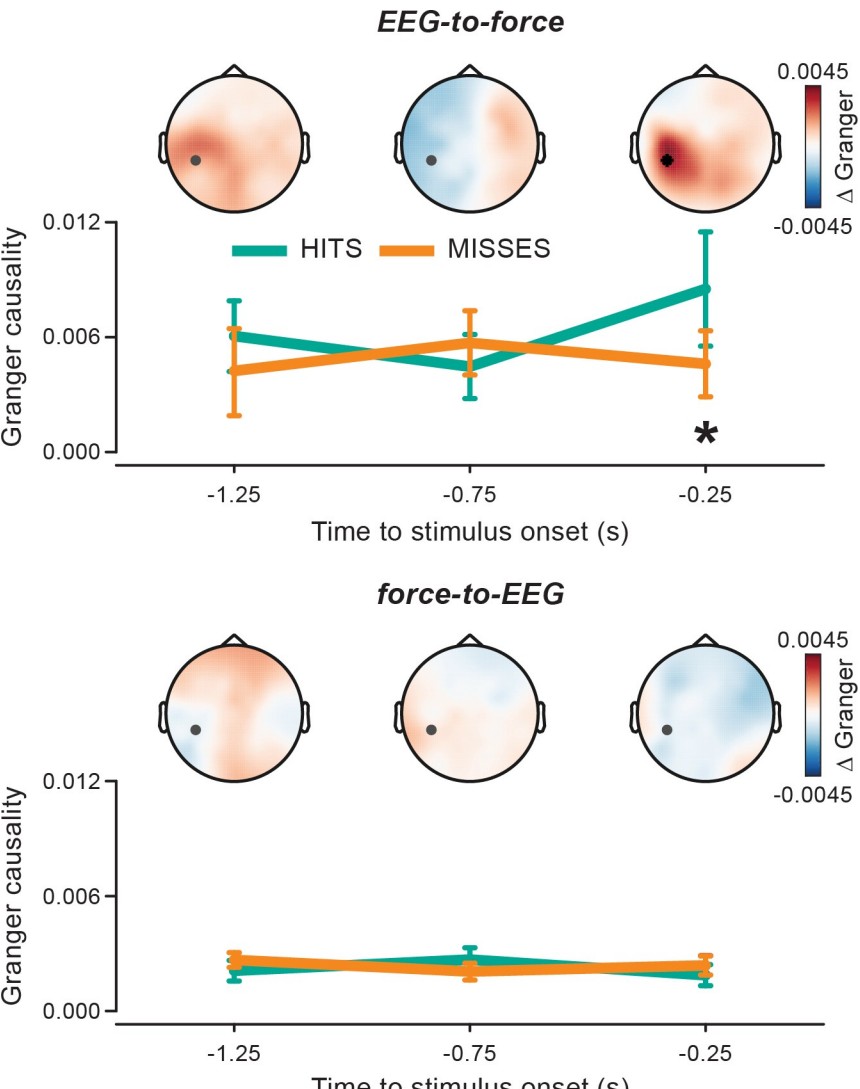

**Fig 7. EEG-to-force alpha Granger causality predicts perception just before stimulus onset.** Granger causality in the EEG-to-force (top) and force-to-EEG (bottom) directions evaluated at frequency 10.5 Hz and electrode CP5 (marked in gray in the topographic maps) is shown for hits and misses as a function of time before stimulus onset (i.e., for 3 nonoverlapping prestimulus 0.5-second time windows centered at −1.25, −0.75, and −0.25 seconds). Topographies show the hits–misses difference in Granger causality at corresponding times (the black asterisk indicates that electrode CP5 survived permutation statistics for the hits–misses contrast; $p = 0.0158$). For the underlying data, see https://osf.io/VSZCB/. EEG, electroencephalography.

findings [43–45], prestimulus alpha power tends to be lower over posterior electrodes for hits compared with misses (albeit the effect does not reach statistical significance; cluster $p > 0.05$; S3 Fig). Given that the higher-power condition (misses) is here associated with lower coherence, potential confounds fall within the second aforementioned case. To deal with this confound, we used an approach based on data stratification. This aimed at equating as much as possible the distributions of occipital alpha power for hits- and misses-trials by means of a random subsampling procedure (see S3 Fig for more details). Results for the stratified data confirm the original pattern: Coherence computed on the entire prestimulus window (same as shown in Fig 3B) as well as at time −0.16 seconds and lag −0.2 seconds (same as shown in Fig

6) is significantly stronger for hits than for misses (S3 Fig). These analyses clearly rule out that the observed coherence modulation is explained by prestimulus modulations of alpha power due to visual attention and/or ongoing excitability.

Finally, we looked at alpha cortico-force coherence in the poststimulus window. Alpha coherence appears to conserve nearly identical spectral, lag, and spatial properties as in the prestimulus window, showing selective increase around negative lags and over left centroparietal electrodes (see S4 Fig; additional <10-Hz zero-lag coherence is visible in hits and likely reflects stimulus-evoked activity and/or phase-resetting phenomena that will not be investigated here). Alpha-lagged coherence does not undergo detection-related modulations after stimulus onset (no difference between hits and misses as evaluated with cluster-based permutation statistics), nor does it differ at any time point from mean prestimulus coherence (averaged over the −0.5 to 0 second interval; cluster-based permutation statistics). These results corroborate the ongoing nature of alpha coherence, with central alpha leading peripheral alpha in time.

## Discussion

Effective motor behavior entails prediction and continuous monitoring of sensory information. Despite its computational bases have been described by several models [46], little is known about how the coupling between motor and sensory functions is realized at the neural level. The present study shows that visual perception is coupled to the ongoing oscillatory dynamics subtending cortico-motor control, suggesting the operation of a task-independent visuomotor loop.

We report 2 novel findings. First, 10-Hz fluctuations in the motor output are phase synchronized (coherent) with a cortical alpha rhythm, which drives (Granger causes) the peripheral rhythm and foreruns it in time by about 0.2 seconds. Second, this nonzero-lag motor synchronization predicts visual performance. Stimulus detection is facilitated when cortico-motor alpha synchronization is enhanced immediately before stimulus onset. This suggests that the actual visual gain is up-regulated at times when upper and lower sensorimotor centers happen to be in the optimal phase relation for their communication. In accordance with an extensive literature (e.g. [32, 36]), we also observed consistent cortico-motor coherence in the beta range (15–30 Hz) which, conversely, peaks around zero lag and bears no relation with visual perception.

### The motor and sensory side of corticospinal coherence

Coherence between cortical and peripheral (muscle/force) activities—also termed cortico-muscular or corticospinal coherence—is typically greatest in the beta-band and has been largely investigated in relation to motor control. Classically viewed as a somatotopically organized propagation of oscillatory signals from the motor cortex to its downstream spinal/muscle targets [41, 47], it is now thought to reflect the information flow within a cortico-peripheral-cortical loop, involving both descending and ascending pathways [32]. Beta-band coherence with muscle activity has been indeed observed in both human and nonhuman primates across an extended sensorimotor network, comprising motor [33, 41] and somatosensory [48] cortices, spinal centers [49], and even proprioceptive afferences to the dorsal spinal roots [50].

Current accounts of cortico-muscular coherence invariably assign it a major role in the integration of motor and somatosensory (especially proprioceptive) signals, being it either for the purpose of keeping the sensorimotor system accurately calibrated [32], maintaining a stable motor output [47, 51–53], strengthening selective sensorimotor connections, such as those

based on muscle synergies [54] or supporting closed-loop sensorimotor control [49]. However, although a (somato-)sensory contribution to cortico-muscular coherence has been postulated [32, 55, 56], its behavioral relevance has so far only been assessed at the motor (e.g., [51, 52, 57]), not at the perceptual, level.

Alongside muscle activity, the overt motor output, i.e., the mechanical result (force/acceleration) of muscle contraction, is dominated by lower-frequency fluctuations around 10 Hz (better known as physiological tremor). Evidence that these are related to central neural activity is sparse [38, 39, 58, 59] and the origin of physiological tremor is debated [35]. Some works have suggested that descending 10-Hz drive may be selectively dampened by phase-cancellation at the spinal level, thereby reducing cortico-motor alpha coupling [60, 61]; others have argued for a primary (if not exclusive) contribution of peripheral factors—e.g., mechanical resonance [62, 63] and stretch reflex [64]—over central neural factors in tremor generation.

Here we show that the peripheral alpha rhythm is (at least partly) explained by a corresponding cortical rhythm. Despite the existing debate, the alpha coherence reported here shows spectral-, spatial-, and lag-selective properties, which are consistent across different data subsets (hits- and misses-trials; see Fig 4) and support as a whole a genuine central basis of tremor. An important novelty of the present findings is that continuous 10-Hz force fluctuations are maximally coherent with 0.2-second backward-shifted EEG alpha activity, indicating cortical phase-modulation of the (forthcoming) motor output (a similar phenomenon is observable for cortico-EMG coherence; see S1 Fig). Interestingly, motor cortical spiking activity in monkeys shows maximal correlation with generated force at an equally long anticipatory lag (i.e., amounting to approximately 0.2 seconds) [65]. The driving role of cortical alpha on the motor output that is (indirectly) inferred from lag-selectivity is further corroborated by directed connectivity estimates (Granger causality), showing an alpha component in the EEG-to-force, but not in the force-to-EEG, direction. Thus, alpha (lagged) connectivity uncovers distinct neural processes that are involved in motor control but operate on a slower timescale as compared with those contributing to (zero-lag) beta connectivity.

Crucially, we show that this slower alpha-band oscillatory dynamics in cortico-motor control predicts visual perception. This is of particular relevance given that motor control is here functionally decoupled from visual processing. Indeed, in the current task, motor control is aimed at achieving a desired proprioceptive state (target force); in other words, it relies only on the closing of a proprioceptive loop as no visual feedback of the exerted force (or of the hand effector) is provided. Thus, the current findings suggest that, despite the task not demanding it, a visuomotor loop might yet be running in the background and cycling at alpha periodicity with consequences on perception.

## Sensory sampling is locked to the internal motor dynamics

Motor contribution to sensory processes has been mainly addressed in the context of active sensing [16, 29–31, 66, 67]. Indeed, some motor behaviors are inherently endowed with a sensory function. Specifically, when effectors (e.g., the eyes) and sensors (e.g., the retina) share the same anatomical substrate, movements' net effect is that of relocating sensors in space, de facto supporting overt sensory information sampling. Motor exploration (e.g., saccades, whisking, sniffing) often displays a rhythmic pattern. Some authors have proposed that this overt motor dynamics could modulate cortical excitability in a predictable manner, boosting sensitivity for the newly acquired inputs [29–31]. Indeed, monkey data show that visual oscillatory dynamics is entrained to the rhythmic alternation between saccades and fixations [16, 17], and recent human behavioral data support the perceptual relevance of these neural modulations [21, 23, 24, 26].

A distinguishing feature of all exploratory behaviors is that the spatiotemporal pattern of the incoming inputs is unavoidably shaped by the overt movements. Yet it was recently shown that visual perception undergoes similar movement-locked rhythmic fluctuations also when preparing movements with the hand [22]. A follow-up experiment revealed that fluctuations in visual perception are explained on a trial-by-trial basis by corresponding brain oscillations phase-locked to action onset [19]. Action-perception coupling mechanisms may thus go beyond the presence of a direct causal link between movements and information sampling. Still, movement can itself be tightly locked to a spatiotemporal reorienting of attention [24, 68, 69]. The present findings extend and partly reshape this idea by showing that sensory sampling is locked also to the continuous issuing of descending motor signals—even when these are aimed at maintaining a stable motor output (i.e., in the absence of an actual movement).

### A "covert" visuomotor loop: Possible mechanisms

Does the reported phenomenon reflect a genuine visuomotor interaction, and in this case, how is this interaction possibly realized?

Alpha oscillatory activity is generally considered a hallmark of sensory and, in particular, visual areas. Despite alpha activity being clearly strongest over occipital sites, its generators are uncertain [70, 71], and they are probably widely distributed and functionally differentiated [72, 73]. A wealth of evidence has shown that alpha activity indexes neuronal excitability and undergoes selective modulations due to attention (e.g., [45, 74, 75]). One can argue that cortico-force coherence in the alpha-band and/or its association with perception may actually reflect nonmotor modulations, perhaps of attentional origin. This argument can be further declined in 2 ways. The first refers to a potential methodological confound: Modulations of cortical alpha power by (spatial/temporal/intermodal) attention, which affect hits and misses to a different degree, may corrupt coherence estimates and eventually produce spurious associations of coherence with perceptual performance. We should first point out that some attentional effects are unlikely in our study. For example, temporal attention cannot play a role, as the visual stimulus was unpredictable (3-second jitter), minimizing anticipatory stimulus-locked modulations of alpha activity. Hits may still be associated with a more effective deployment of sustained spatial attention to the right visual hemifield (where the stimulus is expected), resulting in lateralized prestimulus alpha power modulations [44, 76]; however, no clear left-lateralization of the hits-misses power difference is observed in our data (S3 Fig). The lack of any (positive/negative) covariation between performance in the visual and motor task also weakens the hypothesis of systematic effects due to trial-by-trial changes in arousal or divided attention (Fig 1C and 1D). Anyhow, irrespective of the putative attentional effect that might be at place here, the aforementioned confound is finally ruled out by the fact that the hits–misses difference in coherence survives a stratification procedure whereby the alpha power distributions are matched between the 2 sets of trials (S3 Fig).

The second and more fundamental argument pertains to the possible origin of the cortical alpha activity that ultimately translates into coherent alpha fluctuations of the motor output (the force). Prominent cortico-spinal projections (the most likely anatomical carrier of cortico-motor coherence) are known to originate from several cortical areas beyond M1 and S1 ([77]; premotor cortex [78] and parietal cortex [79]). These areas can, in turn, be influenced by other functionally connected areas, which, by stretching the chain long enough, can include sensory areas up to V1. However, irrespective of its—albeit important—anatomical origin, such an alpha activity is intrinsically a "motor" activity as it maps directly onto a corresponding motor output. In this respect, the causal association between cortical and peripheral alpha activity (see also results based on Granger causality) constitutes per se a highly specific

functional "localizer" of the motor system activity. At this stage, we believe that the most parsimonious hypothesis is therefore that the identified alpha activity originates among those cortical sites that provide direct projections to the lower spinal motor neurons. This hypothesis is further supported by the left (contralateral to the hand effector) and centroparietal topography of alpha coherence. In this respect, it is worth noting that the sensorimotor cortex also features an alpha rhythm that is distinct from the beta rhythm both anatomically (alpha: posterior, beta: anterior—relative to the central sulcus) and functionally [80]. Intriguingly, the hits–misses coherence contrast highlights a significant contribution also from posterior sites (Figs 3B and 5), although this is much less evident immediately before stimulus onset where the effect is maximal (Figs 6 and 7). One possibility is that the dominant (although unrelated) occipital alpha generators partly contaminate sensor-level effect topographies corresponding to hits-versus-misses and this is exacerbated for the more poorly resolved analyses (i.e., over the entire prestimulus epoch; Figs 3B and 5). However, the posterior contribution is preserved also after accounting for occipital alpha power modulations (S3 Fig), making the aforementioned hypothesis less likely. Another possibility is that one (additional) determinant of the hits–misses difference stems from more posterior, parietal visuomotor integration areas (intraparietal sulcus [IPS]), that are known to contribute to the corticospinal tract [79]. Future studies, ideally exploiting neural stimulation techniques (e.g., transcranial magnetic stimulation), would be needed to determine the exact cortical contributions to the alpha activity highlighted herein.

Overall, our finding corroborates the behavioral relevance of neuronal phase synchronization, adding support to its role in effective communication [13]. Previous evidence shows that interareal (gamma) synchronization within the visual system (V1-V4) predicts the speed of motor responses to visual stimulation [81]. Here, we show that synchronization within a motor network (the corticospinal system) predicts visual detection. As no visuomotor transformation is required by the task at hand, this suggests an automatic and covert coupling of visual inputs sampling to the ongoing motor control processes.

## Functional significance for sensorimotor control

Motor behavior commonly relies on many sources of information, which must be integrated in time and space to achieve an efficient control. Beta- and alpha-band cortico-motor coherence may represent the neural signature of multimodal sensorimotor loops operating in parallel at different timescales and to different purposes. Sensorimotor control depends on quickly updated state estimates of the effectors. In line with previous proposals [32, 49], (zero-lag) beta coherence could serve the operation of a proprioceptive-motor loop through which information is fed forward and back at a relatively fast rate.

Actions, in real-world scenarios, require monitoring of task-relevant sensory feedback and also adapting the motor plan to new data coming from all the senses. Incorporating visual, and possibly multimodal, information into the ongoing motor plans would require however longer integration times [2]. Even when not task-relevant, visual information could be (covertly) monitored within a slower control loop, possibly indexed by 0.2-second lagged alpha coherence. Of course, this does not exclude that the strength and/or properties (e.g., lag tuning) of alpha coherence might be modulated in a context-dependent fashion (e.g., by the importance of visual feedback for motor control).

In conclusion, we suggest that the online monitoring of multimodal sensory information is integral to the organization and control of movement and may be indexed by different sensorimotor communication channels (alpha versus beta coherence) running with different lags. Such a lag specificity may stem from the inherent functional role played by proprioceptive

versus exteroceptive/teleceptive signals in tuning the descending command. The former allows a fine and short latency control of muscle activation via direct modulation of spinal reflexive circuitries. The latter is endowed with far less potential to control the details and timing of muscle recruitment in favor of a more general role in guiding actions at higher levels of the cognitive hierarchy.

Oscillatory mechanisms could thus serve to synchronize incoming inputs with descending motor commands/predictions, effectively regulating the information flow within multimodal and multi-timescale sensorimotor loops.

## Methods

### Subjects

Twenty-five healthy participants were recruited to participate in the experiment. Because of the excessive difficulty in the performance of the task (specifically in isometric force control; see next), 5 participants did not complete the experiment but only attended 1 out of the 3 testing days; the data for these participants were not analyzed.

The remaining 20 participants (11 females; age 22.1 ± 3.2 years, mean ± SD) took part in the full experiment. Participants were all naive with respect to the aims of the study and were all paid (EUR€12.5/testing day) for their participation. Participants were right-handed (by self-report) and had normal or corrected-to-normal vision. The study and experimental procedures were approved by the local ethics committee (Comitato Etico della Provincia di Ferrara, approval number: 170592). Participants provided written, informed consent after explanation of the task and experimental procedures, in accordance with the guidelines of the local ethics committee and the Declaration of Helsinki.

### Experimental setup and procedure

Participants sat in a dimly lit room in front of a CRT screen (21 inches, 85 Hz; Sony Trinitron Multiscan-500PS) at a viewing distance of approximately 60 centimeters. They held a dual-axis custom-made isometric joystick with their right hand that allowed measuring hand force continuously along 2 orthogonal axes via 4 load cells. The joystick was securely fixed to a rigid support to avoid displacement and enclosed in a black casing with an aperture in the front to prevent participants from seeing their hand.

Participants performed concurrently a motor task and a visual detection task. The continuous isometric motor task consisted in a wrist abduction to push the joystick's handle towards one's own body with the right hand. The force level that participants were required to exert (i.e., target force) was set at the beginning of the experiment based on the individual maximal voluntary force (MVF; see next).

Each trial was structured as follows. A dark gray fixation cross (size 0.35˚) was first displayed at the center of the screen. Participants were instructed to start exerting force as soon as they felt ready. As participants began pushing the joystick, 4 horizontal bars (left/right and top/bottom of fixation; horizontal/vertical eccentricity: 7.5˚; see Fig 1B) linearly increased their length towards the center of the screen as a function of the applied force, providing online visual feedback of the exerted force. The target force was reached when the left and right bars (top and bottom bars) met at the center of the screen (see Fig 1B). As soon as the participant succeeded in maintaining force within the desired range (target force ± 0.15*target force) for at least 1.5 seconds, the bars stopped changing length and remained precisely aligned to the center of the screen. From this moment onwards (i.e., when the visual feedback of the force was removed), participants were instructed to keep contraction as constant as possible without feedback for 5.5 seconds and, at the same time, pay attention to the appearance of a brief

(0.012 seconds; 1 frame) visual dot with near-threshold contrast (while maintaining central fixation). The dot (size 7' of visual angle) was shown 7.5˚ to the right of fixation in 85% of the trials; in the remaining 15% of the trials (catch trials), no visual stimulus was displayed. Importantly, the stimulus was unpredictable in time, as it appeared at a time that was randomly drawn from a uniform distribution ranging between 1.6 and 4.6 seconds (i.e., 3-second jitter) after fulfillment of the force criterion and consequent visual feedback removal. At the end of the trial (after 5.5 seconds), indicated by a question mark at the center of the screen, participants released hand contraction and reported verbally whether they had seen ("yes" response) or not seen ("no" response) the stimulus.

Prior to the experiment, we estimated first the individual MVF and then the individual visual contrast threshold, i.e., the contrast yielding 50% of "yes" responses (for stimulus-present trials).

## MVF estimation

Participants were asked to apply their maximal force by pushing the joystick handle backward with their right hand in response to a beep (800 Hz, 0.05 seconds) and maintain the same force for 3 seconds (end of interval marked by a second identical beep). This procedure was repeated three times with a 14-second pause in-between repetitions. MVF was estimated as the mean force during the 3-second interval averaged over at least 2 repetitions in which mean force did not differ (across repetitions) by more than 5%. The entire procedure was repeated until this criterion was satisfied.

The lower is the force to be exerted, the more difficult is its fine control and stabilization; at the same time, high forces induce fatigue over the course of the experiment. For these reasons, the target force used for the experiment was set as the minimum force between 10% and 20% of MVF at which the participant was capable of controlling the joystick with no excessive difficulty. Except from 5 subjects who withdrew before completing the experiment, no other subject reported discomfort or fatigue throughout the experiment.

## Visual contrast threshold estimation

Task and trial structure were the same as already described for the main experiment. The contrast of the visual stimulus was changed on a trial-by-trial basis according to the adaptive QUEST algorithm [82]. We ran 60 trials and fitted the obtained data with a cumulative Gaussian function. The contrast threshold was estimated as the mean of the psychometric function.

Because of possible learning effects, the performance level (hit rate, i.e., percentage of "yes" responses for stimulus-present trials) in the main experiment was continuously monitored, and the stimulus contrast was adjusted throughout the experiment to keep performance near threshold. The hit rate was calculated at the end of each block of trials (i.e., 60 trials). The contrast used in the next block of trials was not changed if hit rate was between 45% and 55%. The contrast was decreased/increased by 0.4 dB if hit rate was within 55%–65% or 35%–45%, respectively, by 0.8 dB if it was within 65%–75% or 25%–35%, and by 1.2 dB if it was >75% or <25%.

A photodiode (2.5 × 2.5 centimeters) was placed in the bottom right corner of the monitor and was used to align the data with millisecond accuracy. Specifically, a white square was displayed on the screen at the position of the photodiode (hidden from view) in synchrony with the removal of the force-related visual feedback as well as at the end of the trial.

Both the voltage signal from the photodiode and that from the isometric joystick were recorded by a data acquisition board (USB-1608GX, Measurement Computing; sampling rate, 5,000 Hz).

The presentation of the stimuli and the data acquisition device were controlled via Matlab (The MathWorks, Inc.; https://www.mathworks.com; RRID:SCR_001622) and the Psychophysics Toolbox (http://psychtoolbox.org; RRID:SCR_002881).

## Data collection

Data collection was split in three different testing days (2 hours testing each day). The experiment involved separate blocks of 60 trials each, with few minutes of rest in-between blocks. Participants completed on average a total of 654 ± 53 (mean ± SD) trials.

## EEG and EMG recording

EEG data were recorded continuously during the experiment (except during MVF and visual contrast threshold estimation) with a 64-channel active electrode system (Brain Products GmbH, Gilching, Germany). Electrooculograms (EOGs) were recorded using 4 electrodes from the cap: FT9, FT10, PO9, and PO10 were removed from their original scalp sites and placed at the bilateral outer canthi and below and above the right eye to record horizontal and vertical eye movements, respectively.

All electrodes were online referenced to the left mastoid. The impedance of the electrodes was kept below 15 kΩ. EEG signals were sampled at 1,000 Hz.

Additionally, EMG was collected from a right arm muscle recruited in the radial deviation of the wrist (extensor carpi radialis longus [ECRL]), a synergic muscle for wrist abduction (flexor carpi radialis [FCR]), and a control muscle (biceps brachii brevis [BBB]). Muscles of interest were located via standard palpation procedures and montage was on the muscle belly with an approximately 3-cm interelectrode distance. EMG was recorded with a wireless system (Zerowire EMG, Aurion, Italy), acquired by a CED board (Micro1401; sampling rate, 5,000 Hz) and visualized online with Signal 3.09 (Cambridge Electronic Design, Cambridge, UK; http://ced.co.uk/products/sigovin; RRID:SCR_017081).

The signal from the photodiode was converted in a TTL signal by an Arduino Due board and used to accurately synchronize all acquired data (EEG, EMG, force) and relevant task events.

## Data analysis

Analyses of behavioral and EEG/EMG data were performed with custom-made Matlab code and the FieldTrip toolbox ([83]; http://www.fieldtriptoolbox.org; RRID:SCR_004849].

Data were first temporally aligned to the visual stimulus presentation and then epoched in 2.5-second-long segments from −1.6 to 0.9 seconds (relative to stimulus onset; see Fig 2).

Force traces were visually inspected, and trials were rejected if the force was at any time (during the relevant epoch) less than 20% of the individual target force.

EEG/EMG segmented data were manually checked for bad channels and/or artifacts in the time domain. Trials containing eye movements (saccades, blinks) within a 0.25-second window before stimulus onset were discarded from the analysis. Independent component analysis (ICA) was used to identify and remove residual artifacts in the EEG signal related to eye movements and heartbeat. Bad EEG channels were excluded from the ICA analysis and subsequently interpolated with a distance-weighted nearest-neighbor approach. Fp1, Fp2, AF7, and AF8 were excluded from the analysis because of the greater noise level in a large number of subjects.

Both EMG and force were down-sampled to 1,000 Hz.

## Behavioral analysis

Behavioral performance was evaluated on the force because this was the output variable that the subject was requested to control throughout the task. Force was first low-pass filtered (35-Hz cutoff frequency) and then analyzed separately in the pre- (−1.6 to 0 seconds) and post-stimulus (0–0.85 seconds) windows.

Prestimulus force values were expressed as the percentage change relative to the individual target force ([force−target force]/[target force]*100); these values and their modulus were averaged over time (−1.6 to 0 seconds) to yield an estimate of motor performance accuracy, i.e., force deviation and absolute error, respectively. Within and intertrial force variability were calculated as the SD across time and across trials, respectively. Linear trends were analyzed by submitting (single-trial) prestimulus force (%) to a linear least-squares fitting and deriving the slope of the best-fitting functions (Fig 1D).

For the analysis on the poststimulus window, we first detrended the force based on the −0.5 to 0 seconds prestimulus interval and then applied baseline correction using the −0.05 to 0 seconds prestimulus interval (in accordance with [34]; Fig 1C).

## Spectral analysis and coherence

The frequency content of force and its phase coherence with cortical activity were first analyzed on the entire prestimulus window (−1.6 to 0 seconds; Fig 2A) for frequencies between 5 and 35 Hz (0.5-Hz step) by using Fourier-based analysis combined with the multitaper method ([84]; 3-Hz spectral smoothing; Fig 3].

Lagged coherence was computed by applying short-time Fourier transform on Hanning tapered data segments. In a first analysis (Fig 2B), we used a fixed 0.6-second force data window (from −1.1 to −0.5 seconds) and computed coherence with a 0.6-second EEG data window that was systematically shifted in time (relative to the force signal) from −0.5 seconds (EEG precedes force by 0.5 seconds; i.e., EEG data window: −1.6 to −1 seconds) up to +0.5 seconds (EEG follows force by 0.5 seconds; i.e., EEG data window: −0.6 to 0 seconds) in 10-millisecond steps (Fig 4A).

Next, we computed lagged coherence with a time-resolved approach (Fig 2C). Coherence was computed between a 0.3-second force sliding window that was advanced over the data from −1 to −0.35 seconds (relative to stimulus onset) in 10-millisecond steps and corresponding but time-shifted (relative to the force), 0.3-second EEG windows. To zoom into the phenomenon of interest (i.e., alpha-band cortico-force coherence), this analysis was restricted to frequencies between 5 and 20 Hz and lags from −0.4 to 0.2 seconds (Fig 4C, 4D and 4E; see S2 Fig for the same analysis performed in the frequency range from 15 to 35 Hz).

Finally, we contrasted time-resolved lagged coherence for hits- and misses-trials. Based on the previous analyses, we selected the frequency showing maximal coherence over CP5 on all trials. Coherence was then computed for the previously selected frequency (i.e., 10.5 Hz) and for lags from −0.4 to 0.2 seconds using a 3-cycle (i.e., approximately 0.286-second) sliding window advanced over the data from −1 to 0.15 seconds (Fig 6).

Cross-correlation of EEG and force (normalized so that the autocorrelations at zero lag are identically 1) was computed on 1-second data segments (from −1 to 0 seconds) that were previously band-pass filtered in the beta (20–30 Hz) and alpha (8–12 Hz) range with a 2-pass Butterworth filter (zero-phase filtering, second order for each single pass; Fig 4B).

## Granger causality

Prior to the computation of Granger causality, we removed the power line interference by estimating and subtracting the 50-, 100-, and 150-Hz components in the EEG data, using a discrete Fourier transform on the entire available data segments (−1.6 to 0.9 seconds). Linear

trends were also removed (method of least squares) from both the EEG and force prestimulus data (–1.6 to 0 seconds).

Frequency-domain Granger causality was then estimated with a nonparametric method: spectral matrix factorization of the cross-spectral density matrix, which was obtained by a Fourier analysis on the entire prestimulus window (–1.6 to 0 seconds; frequencies: 5–35 Hz; Fig 5) or on 3 nonoverlapping 0.5-second windows (centered at –1.25, –0.75, and –0.25 seconds; frequency: 10.5 Hz; Fig 7), using multitapering (3-Hz spectral smoothing). Statistical contrasts (see next section for more details) between hits- and misses-trials were performed separately on Granger connectivity estimated in the EEG-to-force and force-to-EEG directions. Based on the analysis on coherence, contrasts were restricted to the alpha range (8–12 Hz; Fig 5) and, for the time-resolved analysis, to frequency 10.5 Hz and electrode CP5 (Fig 7).

## Statistical analysis

Motor performance (accuracy and variability) was compared between hits- and misses-trials using conventional paired samples $t$ tests.

All other statistical comparisons between hits- and misses-trials were performed using cluster-based permutation tests [85] that allow us to deal more effectively with the multiple comparisons along the spatial, frequency, and time dimensions. According to this nonparametric statistical approach, all samples exceeding an a priori decided threshold (uncorrected $p < 0.05$, 2-tailed) for univariate statistical testing (dependent-sample $t$ test) are selected and subsequently clustered on the basis of their contiguity along the relevant (spatial, spectral, and temporal) dimension(s). Cluster-level statistics is computed by taking the sum of $t$-values in each cluster. This sum is then used as test statistic and evaluated against a surrogate distribution of maximum cluster $t$-values obtained after permuting data across conditions (at the level of participant specific condition averages). To generate the surrogate distribution, we used 5,000 permutations. The $p$-value is given by the proportion of random permutations that yields a larger test statistic compared with that computed for the original data.

## Supporting information

**S1 Fig. Cortico-EMG coherence.** Lag- and frequency-resolved coherence between EEG and EMG activity recorded from a wrist extensor muscle (ECRL). Lagged cortico-EMG coherence is calculated on 0.6-second data windows in the same way as shown in Fig 4A for cortico-force coherence. The raw EMG has been high-pass-filtered at 10 Hz and rectified before computing coherence. Cortico-EMG coherence shows similar properties as those observed for cortico-force coherence. Coherence in the beta-band (20–30 Hz) is symmetrically distributed around zero lag and is maximal over contralateral central electrodes (left topography; C1 marked in gray). Coherence in the alpha-band (9–11 Hz) is biased towards negative lags and distributed over contralateral centroparietal electrodes (right topography; CP5 marked in gray). Notably, alpha-band cortico-EMG coherence peaks at a slightly shorter lag (approximately –0.14 s) compared with its cortico-force counterpart (approximately –0.2 s), which is compatible with the fact that muscular activity anticipates in time its mechanical outcome (i.e., force). ECRL, extensor carpi radialis longus; EEG, electroencephalography; EMG, electromyography. (TIF)

**S2 Fig. Lag, spectral, and spatial properties of beta-band cortico-force coherence.** (A) Lag-frequency coherence representation as in Fig 4C but computed on frequencies between 15 and 35 Hz and evaluated at electrode C1 (where coherence in the beta range is maximal). (B) Lag (left) and spectral (right) tuning of beta cortico-force coherence expressed as the relative

percentage change in coherence averaged over frequencies between 20 and 30 Hz and lags between −0.08 and +0.08 seconds (i.e., lag of max. beta coherence on all trials [0 seconds] ±1 SD across subjects), respectively. (C) Topographies show coherence at frequency 25 Hz and lag zero for all trials (top), hits (middle), and misses (bottom).
(TIF)

**S3 Fig. Prestimulus modulations in EEG alpha power do not account for cortico-force coherence modulations.** (A) Time-frequency plot of power modulations (hits/misses) averaged over selected occipital EEG electrodes (marked in gray in the topographic map). The topography shows the hits-misses difference in power calculated at stimulus onset (0 s) and at frequency 10.5 Hz. (B) Nonlagged (left; same as in Fig 3) as well as lagged and time-resolved (right, same as in Fig 6) coherence analysis performed on data stratified on occipital alpha power. Specifically, for the stratification we used the power at frequency 10.5 Hz in the relevant time window (left: 1.6-second prestimulus; right: 0.3-second windows from −1 to 0 seconds), averaged over selected occipital channels where the hits-misses difference is maximal (Oz, O1, O2, POz, PO3, PO4; highlighted in gray in A). The power distributions were compiled for hits and misses and then binned in 10 equally spaced bins. The number of trials in each bin for hits and misses was then equated by a random subsampling procedure which aims at matching as much as possible the parameter averages (as implemented in Fieldtrip, function: ft_stratify, method: 'histogram', 'equalbinavg'). This procedure led on average to the removal of 73.05 ± 5 and 76.55 ± 4.4 trials per subject (mean ± SEM) for the first and second analysis, respectively. The difference in coherence between hits and misses after data stratification has then been evaluated by means of a one-tailed cluster-based permutation test at frequency 10.5 Hz and, for the second analysis, at time point −0.16 seconds and lag −0.2 seconds. For both analyses, we obtain significant results: black asterisks mark the electrodes that survived cluster-based permutation test for the hits-misses contrast. EEG, electroencephalography.
(TIF)

**S4 Fig. Poststimulus alpha cortico-force coherence.** Lag-frequency coherence representation as in Fig 4C but averaged over the poststimulus period (0–0.5 seconds) for all trials as well as separately for hits- and misses-trials. Topographies show corresponding poststimulus coherence (spatially z-scored before averaging across subjects) at 10.5 Hz and lag −0.2 seconds for all trial categories.
(TIF)

## Author Contributions

**Conceptualization:** Alice Tomassini, Alessandro D'Ausilio.

**Formal analysis:** Alice Tomassini.

**Funding acquisition:** Luciano Fadiga, Alessandro D'Ausilio.

**Investigation:** Alice Tomassini, Pauline Hilt.

**Methodology:** Alice Tomassini, Eric Maris.

**Project administration:** Alessandro D'Ausilio.

**Supervision:** Alessandro D'Ausilio.

**Writing – original draft:** Alice Tomassini.

**Writing – review & editing:** Alice Tomassini, Eric Maris, Pauline Hilt, Luciano Fadiga, Alessandro D'Ausilio.

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
