## [Editor Report · Decision Letter 0]

29 Jun 2020

Dear Dr Tomassini, 

Thank you for submitting your revised manuscript entitled "Visual sampling is locked to the internal dynamics of cortico-motor control" for consideration as a Research Article by PLOS Biology.

Your revision and response to reviewers have now been evaluated by the PLOS Biology editorial staff, as well as by the original Academic Editor, and I am writing to let you know that we would like to send your submission out for external peer review.

Please re-submit your manuscript within two working days, i.e. by Jul 01 2020 11:59PM.

Kind regards,

Gabriel Gasque, Ph.D.,

Senior Editor

PLOS Biology

---

## [Decision Letter · Decision Letter 1]

13 Aug 2020

Dear Dr Tomassini,

Thank you for submitting your revised Research Article entitled "Visual sampling is locked to the internal dynamics of cortico-motor control" for publication in PLOS Biology. I have now obtained advice from the original reviewers 2-4 and have discussed their comments with the Academic Editor. The Academic Editor also assessed the way you responded to the concerns originally raised by reviewer 1, who couldn't re-review.

We're delighted to let you know that we're now editorially satisfied with your manuscript. However, based on the feedback provided by reviewer 3, we would like you to change the title to "Visual detection..." rather than "Visual sampling..."

In addition, before we can formally accept your paper and consider it "in press", we also need to ensure that your article conforms to our guidelines. A member of our team will be in touch shortly with a set of requests. As we can't proceed until these requirements are met, your swift response will help prevent delays to publication. Please also make sure to address the data and other policy-related requests noted at the end of this email.

*Copyediting*

*Published Peer Review History*

*Early Version*

*Submitting Your Revision*

Sincerely,

Gabriel Gasque, Ph.D.,

Senior Editor,

ggasque@plos.org,

PLOS Biology

DATA POLICY:

Unfortunately, we were not able to access your Dyad link: 10.5061/dryad.vq83bk3ps

Note that we do not require all raw data. Rather, we ask for all individual quantitative observations that underlie the data summarized in the figures and results of your paper. For an example see here: http://www.plosbiology.org/article/info%3Adoi%2F10.1371%2Fjournal.pbio.1001908#s5

These data can be made available in one of the following forms:

Regardless of the method selected, please ensure that you provide the individual numerical values that underlie the summary data displayed in the following figure panels: Figures 1CD, 3AB, 4D, 5A, 6, 7, and S2B. 

Please also ensure that each figure legend in your manuscript include information on where the underlying data can be found and ensure your supplemental data file/s has a legend.

Reviewer remarks:

Reviewer #2: While as the authors acknowledge, the results are correlational, the new analyses, mainly in response to reviewer 1's questions do provide additional and, in my opinion, convincing evidence for an alpha wavelength visuo-motor loop. I have no more questions or concerns. 

Reviewer #3: While the added analyses (granger and stratification) are very helpful and convincing, I have a suggestion concerning the new title, which I find misleading. It suggests a task with proven (or at least calling for) temporally modulated visual processing (sampling), however, the task used is best solved with continuous visual processing. While of course, visual processing is not continuous, using visual sampling as such in the title feels suboptimal. I would suggest a more neutral phrasing like „Visual processing is locked to the …"

Reviewer #4: In the revised manuscript, the authors have addressed my concerns. As a result, the manuscript has significantly improved. I appreciated the authors' efforts.

---

## [Editor Report · Decision Letter 2]

14 Sep 2020

Dear Dr Tomassini,

On behalf of my colleagues and the Academic Editor, Alexander Gail, I am pleased to inform you that we will be delighted to publish your Research Article in PLOS Biology. 

Early Version

PRESS 

Kind regards,

Vita Usova

Publication Assistant, 

PLOS Biology

on behalf of

Gabriel Gasque,

Senior Editor

PLOS Biology